# Switching between the Forest and the Trees: The Contribution of Global to Local Switching to Spatial Constructional Abilities in Typically Developing Children

**DOI:** 10.3390/brainsci10120955

**Published:** 2020-12-09

**Authors:** Isa Zappullo, Luigi Trojano, Roberta Cecere, Gennaro Raimo, Monica Positano, Massimiliano Conson

**Affiliations:** Department of Psychology, University of Campania Luigi Vanvitelli, 81100 Caserta, Italy; isa.zappullo@unicampania.it (I.Z.); luigi.trojano@unicampania.it (L.T.); roberta.cecere@hotmail.com (R.C.); gennaro.raimo.93@gmail.com (G.R.); positanomonica@hotmail.it (M.P.)

**Keywords:** global/local perception, visual processing, visuospatial abilities, drawing, Rey-Osterrieth Complex Figure, block assembly, Block Design test

## Abstract

Background: Spatial analysis encompasses the ability to perceive the visual world by arranging the local elements (“the trees”) into a coherent global configuration (“the forest”). During childhood, this ability gradually switches from a local to a global precedence, which contributes to changes in children’s spatial construction abilities, such as drawing or building blocks. At present, it is not clear whether enhanced global or local processing or, alternatively, whether switching between these two levels best accounts for children’s spatial constructional abilities. Methods: We assessed typically developing children 7 to 8 years old on a global/local switching task and on two widely used spatial construction tasks (the Rey–Osterrieth Complex Figure and the Block Design test). Results: The ability to switch from global to local level, rather than a global or a local advantage, best accounted for children’s performance on both spatial construction tasks. Conclusions: The present findings contribute to elucidate the relationship between visual perception and spatial construction in children showing that the ease with which children switch perception from global to local processing is an important factor in their performance on tasks requiring complex drawing and block assembling.

## 1. Introduction

Spatial construction refers to the ability to reproduce spatial arrays by organizing details into an integrated spatial configuration, as in complex figure drawing and block assembling [1,2]. Studies on typically developing children have demonstrated that spatial construction involves many abilities such as spatial analysis, spatial manipulation, visual-motor coordination, executive functions and verbal skills [3,4,5,6,7,8,9,10,11]. In particular, much of the data highlight the contribution of spatial analysis to spatial construction performance [1,2,5,6,8,9].

Spatial analysis encompasses the ability to perceive the visual world segmenting it into a set of constituent parts (local processing) and integrating these parts into a coherent whole (global processing) [1,2]. Classical studies on global/local processing have shown a global bias in neurotypical adults, whereby they respond faster and more accurately to global rather than local information [12,13]. However, this global precedence (the “forest before trees”; Ref. [12]) emerges slowly during the process of development. Indeed, a gradual shift from local to global processing seems to take place from about 6 years of age onwards [14,15,16], with the global bias typically becoming progressively refined over a period that extends well into adolescence [17].

It has been suggested that the progressive maturation of spatial analysis plays an important role in the developmental changes observed in spatial constructional performance in school-age children [1,2]. For instance, Martens et al. [18] investigated organizational strategies used by children aged 5–7 years to reproduce the Rey–Osterreith Complex Figure [19,20], and demonstrated that 5-year-olds use more local strategies that are focused on the details, whereas from 7 years onwards, children rely more on a global approach. This developmental pattern confirms the classical evidence presented by Akshoomoff and Stiles [21] showing a gradual shift from local to global strategies to reproduce the Rey–Osterreith Complex Figure in children aged 6–9 years. Thus, these findings suggest that the progressive enhancement of global processing allows the gradual refinement of spatial constructional skills throughout the developmental process. However, data obtained from studies on atypical development have shown that the stronger the local perception, the better the spatial constructional performance. In particular, studies on children with autism spectrum disorders have consistently demonstrated that enhanced local processing results in better performance compared to typical controls on different kinds of spatial constructional tasks, such as realistic drawing [22,23] and block assembly [24]. Additionally, a study on art students suggested that perceptual flexibility can predict drawing skills. Art students who showed superior drawing skills (i.e., more realistic drawing) compared to non-art students were also better at switching between global and local levels of a visual stimulus [25] or between different aspects of an ambiguous figure [26].

Taken together, it is far from clear how global and local processing contributed to the spatial constructional abilities of typically developing children, and whether enhanced global or local processing or, alternatively, switching from global to local or vice versa, enhanced children’s performance on spatial construction tasks.

In the present study, we aimed to examine these alternatives by testing the effect of global and local perception, and of switching between these two levels of analysis, on the performance of typically developing children on the Rey–Osterrieth Complex Figure [19,20] and the Block Design test [27], the most widely used tasks to assess spatial construction [1,2,28,29]. Global and local perception were assessed by means of a modified version of a Navon hierarchical figures task [12], i.e., the global/local switching task, that allowed us to separately measure global processing, local processing, switching from global to local processing and vice versa [25].

Following the literature on sex differences in spatial abilities [8,9,30,31,32], we also tested whether sex differences in global/local processing might have an impact on spatial constructional performance. Importantly, moreover, in the present study we recruited children within the age range of 7 to 8 years because, as discussed above, the available evidence suggests that a transition starts to occur from local to global processing [14,15,16] from 6 years onwards; however, there are individual differences as some children continue to rely on a local strategy while others rely more consistently on global processing [33,34]. Thus, by employing the global/local switching task we were able to assess the way in which individual differences in switching abilities, global perception and local perception influenced spatial constructional performance.

## 2. Materials and Methods

### 2.1. Participants

The sample was recruited from second grade classes of primary schools in the Campania region in Southern Italy. Participants were included in the study only if the following inclusion criteria were met: (i) typical cognitive development, as expressed by a score higher than the 15th percentile for the Raven’s Coloured Progressive Matrices test [35]; (ii) no clinical diagnosis of neurologic, neuropsychiatric or neurodevelopmental disorders, as reported by parents. Ninety-one children (52 females and 39 males) fulfilled the inclusion criteria. The sample had a mean age of 7.7 years (SD = 0.3; range: 7–8 years). All participants were right-handed and spoke Italian as their native language; their families had a mean score of 27.91 (SD = 14.78 range: 6–66) on a validated questionnaire for determining socioeconomic status (SES; [36,37]).

The study was conducted in accordance with the Declaration of Helsinki and approved by the Local Ethics Committee—Department of Psychology, University of Campania Luigi Vanvitelli (Ethical approval code N:34/2020); written informed consent was obtained from all parents of the participants prior to testing.

### 2.2. Tasks

#### 2.2.1. Global/Local Switching Task

Participants underwent a modified version of the hierarchical Navon figures task [25]. The stimuli were presented as white shapes on a black background and consisted of a large geometric figure made up of small local geometric figures. The global shapes covered at a visual angle of approximately 7° along the widest axis (at a viewing distance of 60 cm) and were presented one at a time at one of four positions (left-up, right-up, left-down and right-down) around a fixation point at the center of a computer screen. Each global shape was composed of approximately 18 local shapes (0.9° visual field, at a viewing distance of 60 cm), which were spatially organized to create the global figure.

Participants indicated whether a circle or square was present in the displayed figure, which could be either at the global (large shape) or local (small shape) level. They pressed, as fast and accurately as possible on a QWERTY keyboard, the “b” key if a circle was present or the “h” key if a square was present. Importantly, by presenting circles and squares at the global (large shape) or local (small shape) level, an implicit attentional switch between global and local processing levels in the participants’ visual perception was induced.

The trial sequence and the experimental conditions are presented in Figure 1. Each trial started with a fixation cross (600 msec) followed by the stimulus, which remained on the screen until a key was pressed. Each participant completed 128 trials: 4 stimuli (large circle/small diamonds; large square/small triangles; large triangle/small circles; large diamond/small squares) × 4 positions (left-up, right-up, left-down and right-down) × 8 repetitions. The stimuli presentation order was arranged to obtain the following combinations of stimulus pairs: (i) LL (a local trial followed by a local trial); (ii) GG (a global trial followed by a global trial); (iii) GL (a global trial followed by a local trial); (iv) LG (a local trial followed by a global trial).

The experiment thus included 16 pairs for each combination. In congruent pairs (GG or LL), circles or squares were presented at the same level of spatial analysis (local or global), whereas in incongruent pairs (GL or LG), circles or squares were presented at different levels of spatial analysis (Figure 1). Thus, a 2 (level: global or local) × 2 (congruency: congruent or incongruent pairs) factorial design was tested. Eight practice trials were given and discarded from the analysis.

Stimuli randomization, presentation and data collection (accuracy and RTs) were performed using OpenSesame 3.3.0 software (http://www.cogsci.nl/opensesame).

#### 2.2.2. Rey–Osterrieth Complex Figure

Drawing skills were evaluated by means of copying the Rey–Osterrieth Complex Figure [19,20,38]. Participants were presented a black-and-white complex drawing comprising 18 geometrical elements. The task required participants to copy the figure accurately, without any time restriction. Accuracy scores were assigned adopting Rey’s [39] scoring system, which identifies 18 basic elements in the figure. Reproduction of each graphic element was rated on a scale of 0–2 points: 2 points when the element was completely and properly placed; 1 point when it was incomplete but properly placed, or when complete but poorly placed; 0.5 points when the element was incomplete and poorly placed but recognizable; 0 points when it was absent or not recognizable (score range: 0–36).

#### 2.2.3. Block Design

Building blocks were evaluated by means of the Block Design subtest from the Italian adaptation of the Wechsler Intelligence Scale for Children, fourth edition (WISC-IV; Ref. [27,40]). The task required participants to observe two-/three-dimensional models and use red and white blocks to reproduce them within a time limit. The task was composed of 14 items, and the total time needed to solve each item was recorded. Items had a different scoring system. For the first three items, each correct response at the first attempt and within the time limit was scored 2, while the second presentation was scored 1. For items 4 to 8, each correct response within the time limit was scored 4, while for the items 9 to 14 each correct response was scored as a function of the time taken to respond: 7 points for the time range 1–30 s, 6 points for the time range 31–50 s, 5 points for the time range 51–70 s and 4 points for the time range 71–120 s. The test was stopped after three consecutive responses credited with a score of 0 (score range: 0–68).

### 2.3. Procedure

Each participant was individually tested in a quiet room at school over a single session, lasting about 30 min. The administration order of the measures was randomized across participants.

### 2.4. Statistical Analysis

To test whether the spatial constructional performance could be accounted for by enhanced global or local processing or by switching between the two processing levels, multiple regressions were performed on spatial constructional measures. In particular, two stepwise multiple regressions were run on the Rey–Osterrieth Complex Figure score (the first model) and on the Block Design score (the second model). Independent variables including sex, SES, switching from global to local (GL) and from local to global (LG), global processing (GG) and local processing (LL) were entered in a stepwise fashion, and accuracy and RTs were entered separately.

We also provided data on the participants’ performance on the global/local switching task and the two spatial construction tasks as a function of sex. In particular, statistical analysis of the global/local switching task took into account only responses to the second trial of each pair. Two 2 × 2 × 2 mixed ANOVA designs were performed separately on the accuracy and RTs for correct trials, with congruency (congruent and incongruent trials) and processing level (global and local) as within-subject factors, and with participants’ sex (females and males) as a between-subject factor. Bonferroni-corrected post-hoc comparisons were performed when necessary. Before analyzing data, trials outside the range of 150–3600 ms were discarded from the analysis (in total, 5824 s trials were recorded, and 93% of the total number of trials was included in the analysis; total loss, 7% of trials); one male participant was also discarded from the analysis since his overall accuracy was well below 3.5 SD from the mean.

All analyses were performed using the Statistical Package for Social Sciences (SPSS Inc, version 22.0; IBM Corp., Armonk, NY, USA). Raw data are available upon request to the first author (I.Z.).

## 3. Results

### 3.1. Regression Analysis

In the stepwise regression analysis on the Rey–Osterrieth Complex Figure, one model was significant (F(1,88) = 6.37, *p* = 0.013) and revealed that ACC-GL (β = 0.260, t(88) = 2.52, *p* = 0.013) was the only independent and specific predictor of drawing performance (Table 1; Figure 2); no multicollinearity was found (tolerance value = 1.00; variance inflation factor (VIF) = 1.00).

The stepwise regression analysis on the Block Design contained two steps; the first model was significant (F(1,88) = 4.27, *p* = 0.042) and showed that ACC-GL (β = 0.215, t(88) = 2.07, *p* = 0.042) was the only independent and specific predictor of drawing performance; the fit of the model was increased by the second step (R^2^_diff_ = 0.098). The final model was significant (F(2,87) = 5.83, *p* = 0.004) and showed that ACC-GL (β = 0.304, t(87) = 2.87, *p* = 0.005) and RTs-Switch score (β = −0.283, t(87) = −2.66, *p* = 0.009) were independent and specific predictors of performance (Table 2; Figure 2); no multicollinearity was found (tolerance value = 0.900; variance inflation factor (VIF) = 1.111).

### 3.2. Global/Local Switching and Spatial Construction Tasks as a Function of Sex

Mean ACC and RTs on global/local switching task, separately for sex, are reported in Figure 3. Results of ANOVA on accuracy only showed a significant main effect of congruency (F(1,88) = 5.18, *p* = 0.025, η^2^_p_ = 0.056), with higher accuracy in congruent (mean = 0.89, S.D. = 0.13) than incongruent (mean = 0.87, S.D. = 0.15) conditions. No other main effect or interaction was statistically significant (*p* > 0.05).

Results of ANOVA on RTs showed significant main effects of congruency (F(1,88) = 53.02, *p* < 0.001, η^2^_p_ = 0.376), with faster RTs in congruent (mean = 1338.5, S.D. = 274.5) than incongruent (mean = 1479.1, S.D. = 321.4) conditions, and of sex (F(1,88) = 7.69, *p* = 0.007, η^2^_p_ = 0.080), with faster RTs in males (mean = 1328.9, S.D. = 435.4) than females (mean = 1487.8, S.D. = 372.1). No other main effect or interaction was statistically significant (*p >* 0.05).

Analysis conducted to test possible sex differences on spatial constructional tasks showed that females and males did not significantly differ on both Rey–Osterrieth Complex Figure (females: mean = 20.6, S.D. = 6.2; males: mean = 18.8, S.D. = 6.3; t(88) = 1.49, *p* > 0.05) and Block Design (females: mean = 17.5, S.D. = 8.2; males: mean = 16.2, S.D. = 7.3; t(88) = 0.59, *p* > 0.05).

## 4. Discussion

When testing the role of global/local processing in spatial construction, we found that both figure drawing and block assembling were predicted by switching from the global to local level. The score on Rey–Osterrieth Complex Figure was accounted for by GL switching accuracy only, whereas the Block Design was predicted by both GL switching accuracy and switching speed (RTs). Although the cognitive bases for the two spatial constructional tasks do not fully overlap [29] (see also [41,42,43]), the difference in predictors could be likely ascribed to the different scoring systems used for the two tasks. Here we adopted the classical scoring systems, which are based on reproduction accuracy (without time constraints) for the Rey–Osterrieth Complex Figure [19,20,38], and on time-bound accuracy for the Block Design [27,40], thus possibly explaining the significant association of the latter task with GL switching speed. However, beyond this task-related difference, the finding that switching from the global to local processing level can account for both spatial constructional tasks suggests this visual perception ability could be pivotal for spatial construction performance.

Since the present results are derived from testing a sample within a specific age range, caution is needed when generalizing data to the developing population. Indeed, it has been reported that relevant changes occur in both global/local processing and spatial constructional performance at schooling age [1,2,17,18,21,33]. Thus, future studies should verify the present results by adopting a longitudinal research design. In the same vein, we have to consider the models we tested here on the Rey–Osterrieth Complex Figure and the Block Design accounted for a limited amount of variance; thus, future investigations should place the present results within a comprehensive cognitive model of spatial construction.

Recently, Senese et al. [9] found that copying of the Rey–Osterrieth Complex Figure was influenced in a specific and additive way by visual perception, visual-motor coordination, verbal abilities and age, while it was indirectly related to visual attention, working memory and complex spatial abilities. The authors suggested the direct effect of age on drawing might be explained by a missing variable that could correspond to global/local processing.

The present data supported the idea that global/local processing is relevant within cognitive drawing models, but they underscored that a specific aspect of global/local processing (i.e., switching from global to local level) could represent the most crucial aspect. These results might imply the involvement of inhibitory control in the relationship between switching and constructional performance. Indeed, one study found that efficient inhibitory control allowed a flexible switching from global to local processing, but not vice versa, in adults who had achieved a stable global advantage [44]. The same seems true for children 6 to 9 years who display a better ability to detect targets at the global than local level [45]. Thus, improvement in inhibitory control observed among children 6 to 8 years old [46] could allow a flexible switching from the global to local level, but not vice versa, when a global advantage is almost reached [44,45]. However, it has been shown that children 7 years old, who have not reached a stable global advantage as adults, might need the same cognitive control to inhibit the most salient (either global or local) hierarchical level and select a target that appears at the least salient (either global or local) level [47]. Therefore, the present results could not establish whether global to local switching in spatial constructional performance was strongly related to inhibitory control. This issue calls for a direct investigation, but it is worth remembering here, in a study on perceptual abilities involved in drawing skills for art students, Chamberlain et al. [26] showed that better perceptual switching abilities in art than non-art students were surprisingly related to poorer inhibitory control. The authors suggested that reduced inhibitory control, which has been associated with enhanced creativity [48], coupled with enhanced perceptual flexibility could account for artistic drawing skills. The present data and available literature thus cannot establish whether flexible switching is related to an immature, not yet stable, global dominance, or whether children with better perceptual switching are also better in inhibitory control. However, our results would suggest that global to local switching plays a key role in spatial constructional performance, even though the mechanisms involved in switching between global and local processing levels still need to be clarified.

The present findings, showing that participants were faster in congruent than incongruent conditions, fit classical data [12,13] on neurotypically developing individuals, and they support the view that children may show a global advantage at schooling ages, albeit with large individual differences [33,34,49,50]. Such individual differences seem related to different degrees of maturation in visual areas involving the ventral occipito-temporal cortex and the posterior parietal cortex [33,34]. It is worth underlining that we found significant sex differences for the global/local switching task, with males outperforming females, whereas we did not find sex differences for both spatial constructional tasks. A stronger global advantage in men than in women has consistently been found with hierarchical letter stimuli [51,52], but not for the kind of stimuli used in the present study [15]. As for spatial construction, recent studies with elementary school children did not show sex differences on the Rey–Osterrieth Complex Figure with respect to both overall drawing accuracy [38] and the cognitive basis involved in drawing performance [8,9]. Taken together, these findings seem to be consistent with a recent trend in literature suggesting that, in modern times, sex differences in the visuospatial domain might be weakened compared to the past, likely due to the effect of social experiences such as the equal engagement of females and males in technology [8,9,31,53,54,55,56].

## 5. Conclusions

In conclusion, the present findings shed light on the relationships between global/local processing and spatial construction [22,23,24,25,26,57]. A strength of the present study is that we could concurrently test the contributions of switching and individual differences in global and local processing. This means we could demonstrate, at least in children, spatial constructional skills are more strongly linked to the ease with which children switch from the global to the local aspects of a configuration, rather than to enhanced global or local processing. Further studies are warranted to comprehend the mechanisms involved in flexible global to local switching, especially the role played by the inhibitory control. This would allow us to reach a clearer understanding of the relationships between visual perception and spatial construction, not only in typical but also in atypical development, especially in neurodevelopmental disorders [58,59,60] displaying complex spatial processing impairment profiles.

## Figures and Tables

**Figure 1 brainsci-10-00955-f001:**
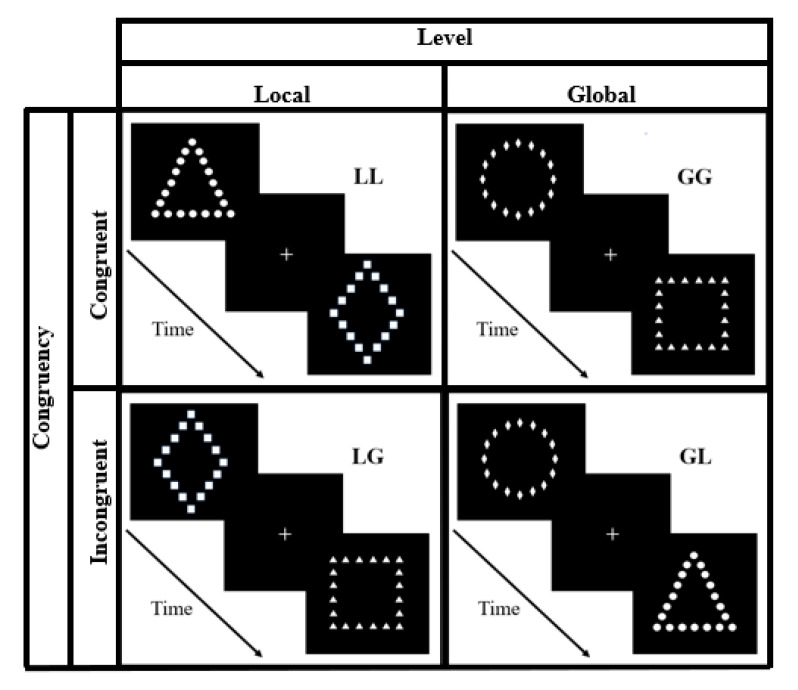
Experimental design. Trial sequence and experimental conditions. Participants were required to indicate whether a circle or a square was present. Target figures were presented at the global (large shape) or local (small shapes) level. Each trial was organized in pairs (local–local, LL; global–global, GG; global–local, GL; local–global, LG). In each pair, circles and squares were at the same (congruent pairs) or opposite (incongruent pairs) level of spatial analysis. As such, each pair fell into a 2 × 2 factorial design for global or local level, congruent and incongruent pairs.

**Figure 2 brainsci-10-00955-f002:**
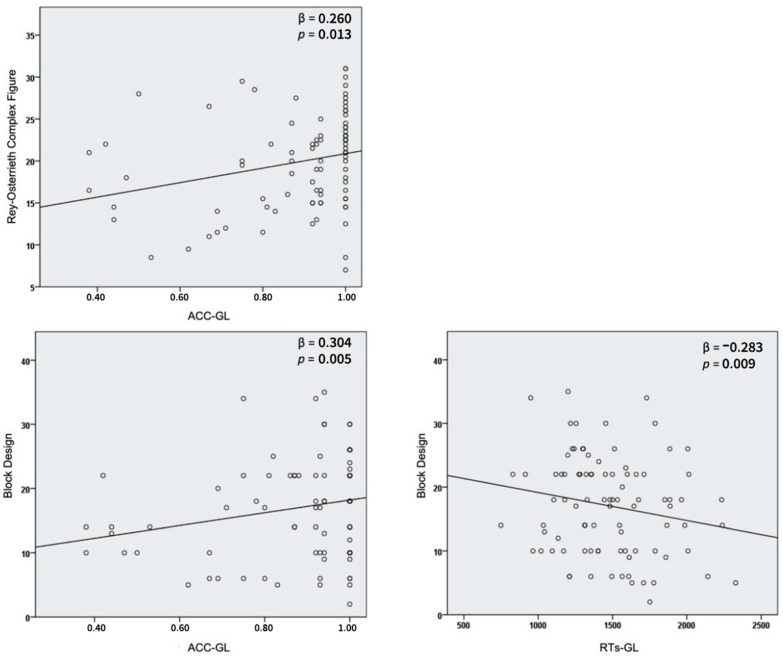
Top left panel: Regression plots of the Rey–Osterrieth Complex Figure with global to local switching accuracy (ACC-GL). Bottom panels: Regression plots of the Block Design with global to local switching accuracy (ACC-GL) and reaction times (RTs-GL).

**Figure 3 brainsci-10-00955-f003:**
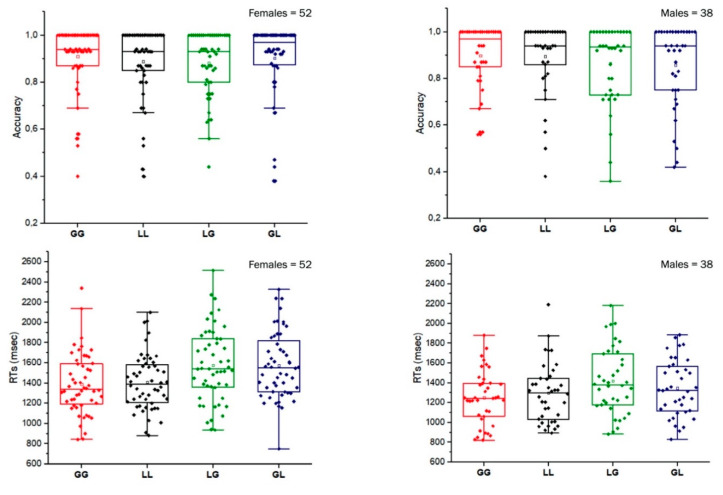
Global/local switching task. Accuracy and RTs, separately for females and males in the four experimental conditions: global–global, GG; local–local, LL; local–global, LG; global–local, GL. Boxes represent 25 and 75 percentiles. The solid line inside the box represents the median of the group, while the empty square in the box represents the mean. Bars above and below the boxes represent the interquartile range. Each individual dot represents a subject.

**Table 1 brainsci-10-00955-t001:** Final regression model on the Rey–Osterrieth Complex Figure.

	B	β	T	*p*
Constant	12.24	-	3.97	0.001
Sex	-	0.131	1.26	0.210
SES	-	0.179	1.73	0.088
ACC-GL *	8.64	0.260	2.52	0.013
ACC-LG	-	0.048	0.41	0.681
ACC-GG	-	0.038	0.32	0.748
ACC-LL	-	0.057	0.39	0.696
RTs-GL	-	−0.058	−0.53	0.597
RTs-LG	-	−0.01	−0.06	0.955
RTs-GG	-	−0.59	−0.54	0.590
RTs-LL	-	−0.080	−0.75	0.458

Sex: participants’ sex (dummy coding: males = 0, females = 1); SES: family socioeconomic status; ACC-GL, global to local switching accuracy: ACC-LG, local to global switching accuracy; ACC-GG, global processing accuracy; ACC-LL, local processing accuracy; RTs-GL, global to local switching speed: RTs-LG, local to global switching speed; RTs-GG, global processing speed; RTs-LL, local processing speed. * Predictors included in final regression model. Fit for final model: F(1,88) = 6.37, *p* = 0.013; R^2^ = 0.067.

**Table 2 brainsci-10-00955-t002:** Final regression model on the Block Design.

	B	Β	T	*p*
Constant	14.49	-	3.04	0.003
Sex	-	0.138	1.29	0.200
SES	-	0.067	0.65	0.518
ACC-GL *	13.99	0.304	2.87	0.005
ACC-LG	-	0.080	0.69	0.491
ACC-GG	-	0.025	0.21	0.831
ACC-LL	-	−0.021	−0.15	0.883
RTs-GL *	−0.007	−0.283	−2.66	0.009
RTs-LG	-	0.092	0.63	0.533
RTs-GG	-	0.016	0.11	0.911
RTs-LL	-	−0.065	−0.47	0.640

Sex: participants’ sex (dummy coding: males = 0, females = 1); SES: family socioeconomic status; ACC-GL, global to local switching accuracy: ACC-LG, local to global switching accuracy; ACC-GG, global processing accuracy; ACC-LL, local processing accuracy; RTs-GL, global to local switching speed: RTs-LG, local to global switching speed; RTs-GG, global processing speed; RTs-LL, local processing speed. * Predictors included in final regression model. Fit for final model: F(2,87) = 5.83, *p* = 0.004; R^2^ = 0.118.

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
