# Peer review of "Switching between the Forest and the Trees: The Contribution of Global to Local Switching to Spatial Constructional Abilities in Typically Developing Children"

_brainsci, 2020, doi:10.3390/brainsci10120955_

Round 1

Reviewer 1 Report

Review of the Ms. “Switching between the forest and the trees: The contribution of flexible visual processing to spatial constructional abilities in typically developing children »

This study investigated the link between flexibility abilities during global/local processing and Rey-Osterrieth Complex Figure and Block Design tests.Results from 7-8 years old children that global/local accuracy switching score was associated to these visuo-spatial tests, whereas reaction times did not reveal significant correlation between the different tasks proposed to children. This topic is interesting. Unfortunately, there are some issues that has to be clarified before a possible publication.

  • Results presented in this study are principally based on correlation analyses. Authors has to be very careful regarding this approach and has to avoid any over or miss-interpretation of the data. In particular, it seems necessary to clearly rule out the possibility that the link bewteen global/local abilities and Rey-Osterrieth/Block Design results are not in fact due to a third variable, for instance to a general inhibitory control or flexibility ability. It seems that the present study mainly insists for the ability to switch between “the forest and the trees”, but it seems also possible that a simple general cognitive control ability – that has nothing to do with global/local abilities – could also explain these results.
  • Page 6, RTs analyses for global/local task as function of sex revealed a significant interaction between congruency and level factors. The description of this interaction is not clearly presented. According to the results mentioned on Table 4, for instance for females, it seems that Global congruent condition was processed faster that Local congruent condition (a traditional behavioral effect during global/local task), but it seems that Global incongruent condition was processed as fast as Local incongruent condition. It explains why this interaction is significant, but this last result is surprising according to the previous literature in the domain. Traditionally, the Local incongruent is largely slowing down by an interference from global irrelevant information, but it seems not the case in the present work. Authors have to clarify and discuss this point.
  • It seems that age influence the present results. Given that only 7-8 years old children with a SD of only 0,3 years participated in this study, I think that age have to be incorporate as a covariable in the analyses, not as a key variable of interest. Authors mentioned this limit regarding the low range of age in the discussion but I think that this analysis have to be removed to avoid any ambiguities for the reader.
  • Significant results are only from accuracies data. Reaction times did not give significant results regarding the main aim of the present work. It seems surprising. Did authors verified that they can perform their analysis with accuracies data? Are there sufficient between subject variance regarding the accuracies? Graphical representation of the results is necessary to provide information regarding variations of individual results.
  • Graphical representations are missing in the paper. It is necessary to clearly present the graphical representations of significant correlations evidenced in the present work, to show the distribution of individual results for the different measures included in the analyses.
  • Minor : typo on Table 4, “GG” have to be presented as “LG” , “LL”, and “GL” conditions

In conclusion, even if the question addressed by this paper seems quite interesting, it seems essential to provide responses to the aforementioned points before a possible publication in Brain Science journal.

Author Response

Point 1. Results presented in this study are principally based on correlation analyses. Authors has to be very careful regarding this approach and has to avoid any over or miss-interpretation of the data. In particular, it seems necessary to clearly rule out the possibility that the link between global/local abilities and Rey-Osterrieth/Block Design results are not in fact due to a third variable, for instance to a general inhibitory control or flexibility ability. It seems that the present study mainly insists for the ability to switch between “the forest and the trees”, but it seems also possible that a simple general cognitive control ability – that has nothing to do with global/local abilities – could also explain these results. 

Response. We thank the Reviewer for the comment on this point. Indeed, we agree that we could not rule out that the role of switching flexibility in spatial constructional performance might imply inhibitory control. Thus, we revised the text, in particular the Discussion section, toning down the emphasis on flexible switching, and underscoring that further studies are warranted to comprehend the mechanisms involved in global to local switching, and especially to clarify the role played by the inhibitory control. It is worth underlining here (please see response to Point 4 for details) that, prompted by the Reviewers’ comments, we revised our statistical analyses aiming at better clarifying whether a specific switching direction (i.e., from global to local or viceversa) could best account from the present results. We, thus, entered in the regression, separately for accuracy and RTs, the four variables from the Global/local switching task (Local-Local, LL; Global-Global, GG; Global-Local, GL; Local-Global, LG). The results showed that the Rey-Osterrieth Complex Figure was predicted by global to local switching accuracy only, whereas Block Design was predicted by both global to local switching accuracy and switching speed (RTs).

Point 2. Page 6, RTs analyses for global/local task as function of sex revealed a significant interaction between congruency and level factors. The description of this interaction is not clearly presented. According to the results mentioned on Table 4, for instance for females, it seems that Global congruent condition was processed faster that Local congruent condition (a traditional behavioral effect during global/local task), but it seems that Global incongruent condition was processed as fast as Local incongruent condition. It explains why this interaction is significant, but this last result is surprising according to the previous literature in the domain. Traditionally, the Local incongruent is largely slowing down by an interference from global irrelevant information, but it seems not the case in the present work. Authors have to clarify and discuss this point.

Response. As better specified below (please see response to Point 4), plotting individual data revealed that overall accuracy of a male participant was well below 3.5 SD from the mean. Thus, we considered him as an outlier and removed him from the two 2 × 2 × 2 mixed ANOVA designs, separately on accuracy and RTs for correct trials, with congruency (congruent and incongruent trials) and processing level (global and local) as within-subject factors, and with participants' sex (females and males) as a between-subject factor. The results of the novel analysis on RTs clearly confirmed significant main effects of congruency, with faster RTs in congruent than incongruent condition, and of sex, with faster RTs in males than females. Results of ANOVA on accuracy showed a significant main effect of congruency, with higher accuracy in congruent than incongruent condition. In the light of these new results, we modified the Discussion section.

Point 3. It seems that age influence the present results. Given that only 7-8 years old children with a SD of only 0,3 years participated in this study, I think that age have to be incorporate as a covariable in the analyses, not as a key variable of interest. Authors mentioned this limit regarding the low range of age in the discussion but I think that this analysis have to be removed to avoid any ambiguities for the reader.

Response. We thank the Reviewer for this comment. We agree with Reviewer and highlighted in the text that we selected a sample with a narrow age range. Thus, we removed the participants’ age from the novel analyses and, consistent with the Reviewer’s expectations, we now obtained much clearer data than before.

Point 4. Significant results are only from accuracies data. Reaction times did not give significant results regarding the main aim of the present work. It seems surprising. Did authors verified that they can perform their analysis with accuracies data? Are there sufficient between subject variance regarding the accuracies? Graphical representation of the results is necessary to provide information regarding variations of individual results.

Response. Prompted by the Referee’s comment, in the revised manuscript we revised our statistical analyses in several respect. First, we revised the variables previously defined as predictors to be regressed on spatial construction performance in order to clarify which aspects of switching could be more related to constructional abilities. In particular, we used measures of the participants’ ability to switch from global to local, and from local to global, and evaluated whether participants were better at processing global or local stimuli. Thus, we entered in the regression, separately for accuracy and RTs, the four variables from the Global/local task (Local-Local, LL; Global-Global, GG; Global-Local, GL; Local-Global, LG). The results showed that the Rey-Osterrieth Complex Figure was predicted by global to local switching accuracy only, whereas Block Design was predicted by both global to local switching accuracy and switching speed (RTs). The difference between the two spatial construction tasks could be likely accounted for by the different scoring systems used for the two tasks. Indeed, in the present study, as in previous classical studies, the ROCF score was based on accuracy only, while the Block Design score was based on time-bound accuracy. Second, as suggested, in the revised manuscript we provided box plots with individual data. This allowed us to identify a male participant whose overall accuracy was well below the 3.5 SD from the mean, thus all the novel analyses were performed after excluding this outlier.

Point 5. Graphical representations are missing in the paper. It is necessary to clearly present the graphical representations of significant correlations evidenced in the present work, to show the distribution of individual results for the different measures included in the analyses.

Response. Following the Reviewer’s comment, in the revised manuscript we added a new figure (Figure 2) providing the significant predictors for the two spatial constructional tasks, and also providing the distribution of the individual data.

Point 6. Minor: typo on Table 4, “GG” have to be presented as “LG”, “LL”, and “GL” conditions.

Response. As reported above, we changed the Table with the new Figure 3.

Reviewer 2 Report

This manuscript is addressing the interesting question how a low level visual processing task (the local and global detection of shapes) is related to the ability to spatial construction ability (as assessed by a demanding copying task and the Wechsler Block Design task) in children of 7-8 years. The authors should be commended for a carefully conducted study using a range of established task in an age group which is not easy to work with. The key finding is that the ability of switching between local and global processing predicts the spatial construction performance. Whilst the result is interesting as such it raises more questions that it is giving answers, possible be related to the fact that a narrow age group was tested. The authors do recognise this issue by pointing to the need for a longitudinal study and discussing a wide range of literature at the end of the paper, which could provide pointers to possible effects of cognitive development.

Not having used the tasks myself, I would have benefited from a little more insight into the experimental methods, I particular the local-global switching task: was there only one response to the stimulus (I might have missed this?): duration of presenting the stimulus sequence (presumably 2 figures with a short break?), one or two key presses?, if one key press, would it ask for circle AND Square, or circle OR Square? – showing cumulative raw data for responses would have helped to disambiguate this…

Also the tables (and/or text) could benefit from a little more detail, such as unit of RT’s (msec?) and other variables, like switch-index or accuracy (units?), etc.

Author Response

Point 1a. Not having used the tasks myself, I would have benefited from a little more insight into the experimental methods. In particular the local-global switching task: was there only one response to the stimulus (I might have missed this?).

Response. As suggested by the Reviewer, in the revised manuscript we reported task instructions to a greater detail in the revised “Global/local switching task” section: “Task instructions required participants to indicate whether a circle or square was present in the displayed figure, which could be either at the global (large shape) or local (small shape) level. They had to press as fast and accurately as possible on the QWERTY keyboard the “b” key if a circle was present or the “h” key if a square was present”.

Point 1b. duration of presenting the stimulus sequence (presumably 2 figures with a short break?), one or two key presses?

Response. We specified in the revised “Global/local switching task” section the following: “Each trial started with a fixation cross (600 msec) followed by the stimulus, which remained on the screen until key press”.

Point 1c. If one key press, would it ask for circle AND Square, or circle OR Square?

Response. Task instructions required participants to indicate whether a circle OR a square was present in the displayed figure. This has been specified in the revised text.

Point 1d. showing cumulative raw data for responses would have helped to disambiguate this.

Response. In the revised manuscript we provided a new Figure (Figure 3) with box plots containing individual data.

Point 1e. Also the tables (and/or text) could benefit from a little more detail, such as unit of RT’s (msec?) and other variables, like switch-index or accuracy (units?), etc.

Response. In the revised manuscript (please also see responses to Reviewer #1), we revised our statistical analyses aiming at better clarifying whether a specific switching direction (i.e., from global to local or viceversa) could best account from the present results. We, thus, entered in the regression, separately for accuracy and RTs, the four variables from the Global/local task (Local-Local, LL; Global-Global, GG; Global-Local, GL; Local-Global, LG). The results showed that the Rey-Osterrieth Complex Figure was predicted by global to local switching accuracy only, whereas Block Design was predicted by both global to local switching accuracy and global to local switching speed (RTs). In the revised manuscript we also provided new figures (2 and 3) making data clearer for the reader (see our response to the comments above).

Round 2

Reviewer 1 Report

Dear Editor, the authors clarified all points raised from my review. The paper is now, to my opinion, ready for publication. Yours fatihfully